# Prototypical Influence Function for Fully Test-Time Adaptation

## Abstract

Test-time adaptation (TTA) addresses domain shift issues in real-world applications. TTA adapts the model considering real-world constraints: (1) TTA does not have access to the training data or the labels of the test data and (2) TTA has limited computational resources for adaptation since it adapts model while performing inference. Due to the constraints, it has been established that model updates based on model-trusting data whose predictions closely aligned with one-hot vectors are effective. Hence, we propose a Prototypical Influence Function (PIF) regularizer utilizing the influence function to assess the influence of adapting a test data point on the loss for model-trusting data. The influence function is impractical for TTA due to computational complexity and the unavailability of model-trusting data. However, by introducing reasonable approximations, we can feasibly use the PIF for TTA. Our experimental results demonstrate consistent performance enhancement when the PIF is applied into the existing TTA methods on various benchmark datasets.

## 1 Introduction

Deep learning models have made significant advancements under the assumption that training and test data are sampled from the same distribution (Krizhevsky et al., 2017). However, in the real world, this assumption is easily violated. For example, there are weather changes such as rain or snow, as well as natural corruptions caused by spots on camera sensors. Conventional deep learning models are vulnerable to such distribution shifts and experience degradation in the presence of natural corruptions (Hendrycks & Dietterich, 2019). Therefore, numerous attempts have been made to robustly address unknown distribution shifts, including domain adaptation (DA) (Csurka, 2017), domain generalization (DG) (Muandet et al., 2013), unsupervised domain adaptation (UDA) (Ganin & Lempitsky, 2015) and source-free domain adaptation (SFDA) (Liang et al., 2020).

The tasks mentioned earlier require access to training (source) data or a separate process to adapt the model to the test (target) data. However, in real-world applications, retrieving source data during inference or allocating considerable time for adaptation is infeasible. These real-world constraints lead to the emergence of the field on test-time adaptation (TTA) (Wang et al., 2021). TTA conducts model adaptation while simultaneously performing inference on test data. It assumes the training data to be inaccessible and solely relies on the pre-trained model with test data that is streamed online. As TTA also lacks access to labels for test data, existing methods often employ objectives such as entropy minimization (Wang et al., 2021) or cross-entropy with pseudo-labels (Goyal et al., 2022). Both of these objectives aim to guide the model's self-supervision, enforcing its predictions. A potential drawback of these approaches is that the objectives may lead the model to collapse in the initial phase of adaptation, when the model has not yet stabilized.

In order to mitigate the performance degradation caused by model collapse, various filtering methods have proposed such as entropy filtering (Niu et al., 2022) or confidence filtering (Sohn et al., 2020). These approaches have experimentally demonstrated that model updates based on reliable samples effectively boost the performance, which implies the importance of the sample selection in label-absent tasks. Therefore, it is crucial to minimize the loss for *model-trusting data*, which refer to samples whose probabilistic outputs obtained from the model closely resemble one-hot vectors, in order to improve model performance in TTA.

We propose a novel regularizer called the prototypical influence function (PIF), derived from the influence function (IF), introduced by Koh & Liang (2017). The IF quantifies how much a data point affects the model parameter without a leave-one-out retraining process. In our work, we reformulate IF tailored to TTA to quantify how much a data point affects the loss of model-trusting data without the adaptation process. The PIF regularizes the model to enhance the positive impact of test data on the model-trusting data, specifically aimed at reducing the loss of model-trusting data. However, there are computationally

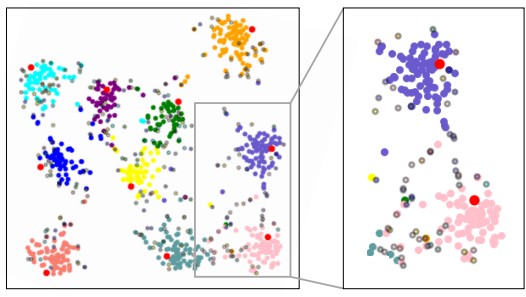

Figure 1: t-SNE visualization of feature embedding of test data and weight prototypes.

intensive terms involved in the IF, making its direct usage practically infeasible. Moreover, obtaining model-trusting data is impossible due to the unavailability of access to the entire test dataset.

To address the first challenge, we employ two approximations to calculate the influence function in a computationally efficient manner. We first obtain an analytical expression for PIF, which can be computed along with a forward pass. Moreover, for the second challenge, we approximate the features of model-trusting data as the weights of the model's last layer. As the last layer of the model is responsible for mapping features to their corresponding classes, model-trusting data align well with the last layer weights that map them to their pseudo-label classes (Tian et al., 2021). Hence, last layer weights effectively represent the class prototypes of model-trusting data in the feature space.

To confirm whether the last layer weights serve as a suitable approximation for the prototypes of model-trusting data, we visualized test samples and the last layer weights in the feature embedding space of the pre-trained model using t-SNE (Van der Maaten & Hinton, 2008). Figure 1 illustrates CIFAR-10-C (Hendrycks & Dietterich, 2019) data with Gaussian noise, where the red circles represent the last layer weights. Colored circles represent test samples whose pseudo-labels match their ground truth labels, while circles with gray edges represent test samples whose pseudo-labels do not match their ground truth labels. Due to the denser clusters of colored circles are constructed around the weights, Figure 1 empirically demonstrates the validity of the weights as the prototypes.

We examined the robustness and effectiveness of the PIF regularizer over public TTA benchmark datasets, including CIFAR-10-C, and ImageNet-C (Hendrycks & Dietterich, 2019). Additionally, we conducted experiments on a large-scale dataset, ImageNet-3DCC (Kar et al., 2022). PIF consistently improves the performance of existing methods across all datasets. We also show that the PIF loss is robust to the value of the hyperparameter. Moreover, we perform comparative experiments against various design choices to analyze the performance of the PIF loss. Our contributions are summarized as follows:

- We introduce the PIF regularizer, which aims to regulate the model by amplifying the positive impact of test data on the loss of model-trusting data.
- We formulate the feasible PIF regularizer which satisfies the limited access and low-resource constraints of TTA.
- We demonstrate the effectiveness of PIF on various benchmark datasets, showing consistent performance improvements while maintaining computational efficiency.

## 2 RELATED WORK

**Test-Time Adaptation (TTA)** TTA is a task that performs inference while adapting to an online test data stream, without access to the source data. It shares a similar paradigm with source-free domain adaptation (SFDA) (Liang et al., 2020), in which that source data is inaccessible. While SFDA performs adaptation before actual inference, TTA requires on-the-fly adaptation during inference time. To meet the tight computation constraints, NORM (Schneider et al., 2020) updates only the batch normalization statistics on mini-batch samples during test time. Furthermore, true labels of the test data are absent in the TTA context, so augmentation methods or unsupervised losses are employed for the TTA task. TTT (Sun et al., 2020), MEMO (Zhang et al., 2022), Test-time Augmentation (Ashukha et al., 2020), and DUA (Mirza et al., 2022) require augmented images of the test data. There are

two main types of unsupervised losses in TTA: the entropy minimization loss and the cross-entropy loss with pseudo-labels. TENT (Wang et al., 2021) and EATA (Niu et al., 2022) optimize entropy minimization loss while adapting batch normalization statistics during test time. PL (Lee et al., 2013) generates pseudo-labels based on model predictions and uses cross-entropy loss to adapt the batch normalization layer parameters to the test data. Also, consistency regularization techniques are used by cross-entropy loss (Chen et al., 2022). Our proposed PIF loss is generic enough to be integrated with different TTA methods.

**Influence Function (IF)** The influence function, introduced by Koh & Liang (2017), estimates how much the model changes when we increase the importance of a training sample. Furthermore, their study also explored the impact of up-weighting a particular training sample on validation loss. Influence function is applied in a variety of tasks, including model debugging, identifying dataset anomalies, and crafting training-set adversarial attacks. In the context of semi-supervised learning, IF is utilized to assign weights to unlabeled data (Ren et al., 2020). IF is also applied in data augmentation (Lee et al., 2020) for quantifying the influence of augmented data. The application of IF extends to tasks such as identifying mislabeled data (Kong et al., 2021) and partial label learning (Gong et al., 2022). Several studies have focused on understanding and analyzing the behavior of IF (Bae et al., 2022; Saunshi et al., 2022). Additionally, efforts have been made to scale up the calculation of IF (Schioppa et al., 2022). In previous works, the IF was used as a tool to understand the relationship between training data, model, and test data, but there were no attempts to use IF values as an objective function. To the best of our knowledge, we propose the first method applying the IF to TTA. We utilize the IF to quantify the relationship between the test data and model-trusting data. Then, the proposed PIF regularizer aims to maximize the IF values.

## 3 PROPOSED METHOD

In the following sections, we propose Prototypical Influence Function (PIF) regularizer for test-time adaptation. We first formalize the problem of TTA (Section 3.1). We quantify the influence of test data on the model-trusting data (Section 3.2). Subsequently, we reformulate the influence function to align with the constraints of TTA. (Section 3.3). Finally, we introduce the PIF regularizer, which enhances the influence of the test data on the loss of model-trusting data (Section 3.4).

### 3.1 PROBLEM DEFINITION

The training data is denoted as $\mathcal{D}_{\text{tr}} = \{(\mathbf{x}_i^s, y_i^s)\}_{i=1}^{N_{\text{train}}}$, where $\mathbf{x}_i^s$ represents the input training samples and $y_i^s$ represents their corresponding labels. The distribution of the training data is denoted as $P(\mathbf{x})$ (i.e., $\mathbf{x}_i^s \sim P(\mathbf{x})$). The training data consists of $c$ classes, where $y_i^s$ takes values from the set $Y = \{1, 2, \ldots, c\}$. On the other hand, the test data is represented as $\mathcal{D}_{\text{te}} = \{\mathbf{x}_j^t\}_{j=1}^{N_{\text{test}}}$ and $\mathbf{x}_j^t \sim Q(\mathbf{x})$. Since test data is corrupted, $P(\mathbf{x}) \neq Q(\mathbf{x})$ and there is no label for test data.

Let's denote the feature extractor as $f_\phi(\cdot)$ and the classifier as $g_{\boldsymbol{w}}(\cdot)$. For the input test data $\mathbf{x}^t$, the feature of the test data can be represented as $f_\phi(\mathbf{x}^t) = \mathbf{h}^t$, and the logit of the test data can be represented as $g_{\boldsymbol{w}}(\mathbf{h}^t) = \mathbf{z}^t$. The prediction of the test data is the softmax of the logit (i.e., $\sigma(\mathbf{z}^t) = \hat{\mathbf{y}}^t$). The set of whole network parameters is represented as $\boldsymbol{\theta}$ which contains the parameters of feature extractor and the parameters of classifier (i.e., $\boldsymbol{\theta} = \phi \cup \boldsymbol{w}$). The classifier consists only of the last fully-connected layer. Then the parameters of classifier $g_{\boldsymbol{w}}(\cdot)$ consists of the bias and weights, namely $\boldsymbol{w} = \{\mathbf{b}, \mathbf{w}_1, \mathbf{w}_2, \cdots, \mathbf{w}_c\}$ where $\mathbf{b}$ denotes the bias.

Lastly, Model trusting data refers to data points with low entropy or high maximum softmax probabilities since TTA is an unsupervised task. Let's denote model-trusting data as $\mathcal{M} = \{\mathbf{x}_k^m\}_{k=1}^M$. The set of model-trusting data is a subset of test data (i.e., $\mathcal{M} \subseteq \mathcal{D}_{\text{te}}$). It is crucial to minimize the loss of model-trusting data in TTA.

Following the prior works (Wang et al., 2021; Niu et al., 2022; Goyal et al., 2022), we update only the parameters of the batch normalization (BN) layer of the feature extractor during adaptation.

### 3.2 Influence of Test Data on the Model-Trusting Data

**Influence of test data on the model parameters**   To understand the impact of adapting the test data to the model, we can consider a change in the model parameters after adapting to test data. By utilizing influence functions, we can assess the impact of an individual test sample on the change in the model parameters without the need for computationally expensive leave-one-out training. We denote the loss of sample $\mathbf{x}_i$ by $\mathcal{L}(\mathbf{x}_i; \boldsymbol{\theta})$. Our optimization objective is based on empirical risk minimization (ERM). Specifically, the empirical risk over $\mathcal{D}_{\mathrm{tr}}$ is defined as $\mathcal{L}(\mathcal{D}_{\mathrm{tr}}; \boldsymbol{\theta}) = \frac{1}{N_{\mathrm{train}}} \sum_{i=1}^{N_{\mathrm{train}}} \mathcal{L}(\mathbf{x}_i^s; \theta)$. Then the optimal parameters of a pre-trained model on the train data can be defined as $\hat{\boldsymbol{\theta}} \overset{\text{def}}{=} \arg\min_{\boldsymbol{\theta} \in \boldsymbol{\Theta}} \mathcal{L}(\mathcal{D}_{\mathrm{tr}}; \boldsymbol{\theta})$. If test data point $\mathbf{x}^t$ added by some infinitesimally small $\epsilon$, the new parameter is given as $\hat{\boldsymbol{\theta}}_{\epsilon, \mathbf{x}^t} \overset{\text{def}}{=} \arg\min_{\boldsymbol{\theta} \in \boldsymbol{\Theta}} \mathcal{L}(\mathcal{D}_{\mathrm{tr}}; \boldsymbol{\theta}) + \epsilon \mathcal{L}(\mathbf{x}^t; \boldsymbol{\theta})$. We can use the following closed-form expression to estimate the change in the model parameters when adapting to $\mathbf{x}^t$ by $\epsilon$. The influence of adding $\mathbf{x}^t$ on the parameters is given by:

$$\mathcal{I}_{\mathrm{add.params}}(\mathbf{x}^t) \overset{\text{def}}{=} \frac{d\hat{\boldsymbol{\theta}}_{\epsilon, \mathbf{x}^t}}{d\epsilon}\bigg|_{\epsilon=0} = -H_{\hat{\boldsymbol{\theta}}}^{-1} \nabla_{\boldsymbol{\theta}} \mathcal{L}(\mathbf{x}^t; \hat{\boldsymbol{\theta}}). \tag{1}$$

where $H_{\hat{\boldsymbol{\theta}}} \overset{\text{def}}{=} \nabla_{\boldsymbol{\theta}}^2 \mathcal{L}(\mathcal{D}_{\mathrm{tr}}; \hat{\boldsymbol{\theta}})$ is the Hessian of the model at parameters $\hat{\boldsymbol{\theta}}$. Detailed proof of Equation 1 is provided in the Appendix.

**Influence of test data on the loss of model-trusting data**   We are interested in understanding the influence of test data on the loss of model-trusting data. This can be measured by the change in the loss of model-trusting data when adapting to the test data. By utilizing the chain rule and Equation 1, we can approximate the influence of adapting test data $\mathbf{x}^t$ on the loss at $\mathbf{x}^m \in \mathcal{M}$ as follows:

$$\mathcal{I}_{\mathrm{add.loss}}(\mathbf{x}^t, \mathbf{x}^m) \overset{\text{def}}{=} \frac{d\mathcal{L}(\mathbf{x}^m; \hat{\boldsymbol{\theta}}_{\epsilon, \mathbf{x}^t})}{d\epsilon}\bigg|_{\epsilon=0} = \nabla_{\boldsymbol{\theta}} \mathcal{L}(\mathbf{x}^m; \hat{\boldsymbol{\theta}})^{\mathsf{T}} \frac{d\hat{\boldsymbol{\theta}}_{\epsilon, \mathbf{x}^t}}{d\epsilon}\bigg|_{\epsilon=0} \tag{2}$$

$$\tag{3}$$

$$= -\nabla_{\boldsymbol{\theta}} \mathcal{L}(\mathbf{x}^m; \hat{\boldsymbol{\theta}})^{\mathsf{T}} H_{\hat{\boldsymbol{\theta}}}^{-1} \nabla_{\boldsymbol{\theta}} \mathcal{L}(\mathbf{x}^t; \hat{\boldsymbol{\theta}}). \tag{4}$$

Equation 4 represents a gradient of $\mathcal{L}(\mathbf{x}^m; \hat{\boldsymbol{\theta}}_{\epsilon, \mathbf{x}^t})$ with respect to $\epsilon$ at nearby $\epsilon = 0$. We can linearly approximate the change in the loss of model-trusting data $\mathbf{x}^m$, after adapting to $\mathbf{x}^t$. The loss change can be demonstrated as follows:

$$\mathcal{L}(\mathbf{x}^m; \hat{\boldsymbol{\theta}}_{\epsilon, \mathbf{x}^t}) - \mathcal{L}(\mathbf{x}^m; \hat{\boldsymbol{\theta}}) \approx \epsilon \times \mathcal{I}_{\mathrm{add.loss}}(\mathbf{x}^t, \mathbf{x}^m). \tag{5}$$

We then estimate the influence of adapting to $\mathbf{x}^t$ on the whole loss of model-trusting data.

$$\mathcal{L}(\mathcal{M}; \hat{\boldsymbol{\theta}}_{\epsilon, \mathbf{x}^t}) - \mathcal{L}(\mathcal{M}; \hat{\boldsymbol{\theta}}) \approx \epsilon' \times \sum_{k=1}^{M} \mathcal{I}_{\mathrm{add.loss}}(\mathbf{x}^t, \mathbf{x}_k^m). \tag{6}$$

Henceforth, we denote by $\mathcal{I}_{\mathrm{add.loss}}(\mathbf{x}^t) = \sum_{k=1}^{M} \mathcal{I}_{\mathrm{add.loss}}(\mathbf{x}^t, \mathbf{x}_k^m)$ the influence of adapting to $\mathbf{x}^t$ on the loss of model-trusting data.

To ensure that the test data $\mathbf{x}^t$ has a positive impact on the loss of the model-trusting data, the loss change should be less than or equal to zero. Since $\epsilon'$ is positive, $\mathcal{I}_{\mathrm{add.loss}}(\mathbf{x}^t)$ should be negative.

### 3.3 Reformulation of Influence Function for TTA

In section 3.2, we quantify the influence of test data on the loss of model-trusting data using the influence function. However, directly calculating $\mathcal{I}_{\mathrm{add.loss}}(\mathbf{x}^t)$ in TTA is not feasible. TTA prioritizes computational efficiency, and the computations required to calculate $\mathcal{I}_{\mathrm{add.loss}}(\mathbf{x}^t)$, such as Hessian inverse calculations and gradients over the entire set of parameters, are computationally expensive for TTA. Furthermore, TTA lacks access to the entire test set and have no label of test data, making it impossible to directly construct model-trusting data. Therefore, to efficiently compute $\mathcal{I}_{\mathrm{add.loss}}(\mathbf{x}^t)$ in TTA, we propose the following two approximations.

**Approximation for influence function**  We employ two approximation strategies, parameter restriction and random projections (Schioppa et al., 2022) to overcome the computational bottleneck and enable efficient implementation of the influence function. Parameter restriction limits the computation to a smaller subset of parameters, usually by selecting only the last layer. Random projections, on the other hand, approximate the Hessian matrix to the identity matrix and reduce influence function to dot products of gradients.

As a result, we obtain the dot product between the last layer gradient of the test data and the model-trusting data which is more practical computation at test time. The inverse Hessian serves the purpose of weighting between two gradients. However, we approximate it with an identity matrix due to the critical efficiency considerations in TTA scenario. To reduce the error arising from this issue and enhance stability, we normalize the gradients before dot product operation. Then Equation 4 becomes cosine similarity between two gradients ($\cos\_\mathrm{sim}(\cdot, \cdot)$). This aspect will be further elaborated upon in the Section 4. Consequently, $\mathcal{I}_{\mathrm{add.loss}}(\mathbf{x}^t, \mathbf{x}_k^m)$ is approximated as follows:

$$\mathcal{I}_{\mathrm{add.loss}}(\mathbf{x}^t, \mathbf{x}_k^m) \approx -\cos\_\mathrm{sim}(\nabla_{\boldsymbol{w}}\mathcal{L}(f_{\boldsymbol{\phi}}(\mathbf{x}_k^m); \boldsymbol{w}), \nabla_{\boldsymbol{w}}\mathcal{L}(f_{\boldsymbol{\phi}}(\mathbf{x}^t); \boldsymbol{w})). \tag{7}$$

**Approximation for class prototype**  Since we need gradients for the last layer to calculate Equation 7, we need to know the value of $f_{\boldsymbol{\phi}}(\mathbf{x}_k^m)$, which is the feature of the model-trusting data. However, since we don't have acces to entire test data and the label of test data, we have to approximate feature of the model-trusting data. Model-trusting data exhibits predictions that are close to one-hot vector, and the last fully-connected layer serves to mapping features to each class (Snell et al., 2017). Consequently, model-trusting data aligns well with the weight prototypes corresponding to each pseudo-label. Therefore, we replace $\mathcal{M} = \{\mathbf{x}_k^m\}_{k=1}^M$ with $\{\mathbf{w}_1, \mathbf{w}_2, \cdots, \mathbf{w}_c\}$. Then the $\mathcal{I}_{\mathrm{add.loss}}(\mathbf{x}^t)$ becomes as follows:

$$\mathcal{I}_{\mathrm{add.loss}}(\mathbf{x}^t) = \sum_{k=1}^M \mathcal{I}_{\mathrm{add.loss}}(\mathbf{x}^t, \mathbf{x}_k^m) \approx \sum_{i=1}^c q(\mathbf{x}^t)(i) \times \mathcal{I}_{\mathrm{add.loss}}(\mathbf{x}^t, \mathbf{w}_i). \tag{8}$$

where $q(\mathbf{x}^t)$ determines how much weighting is applied to each weight prototype in order to calculate their influence and $(i)$ means $i$-th output of the vector. There are three choices for $q(\mathbf{x}^t)$: "Hard weighting", "Soft weighting", and "Uniform weighting". Hard weighting involves calculating influence only for the weight prototype corresponding to pseudo-label of $\mathbf{x}^t$ (i.e., $\tilde{y}^t = \mathrm{argmax}_i(\hat{\mathbf{y}}^t(i))$). In contrast, soft weighting utilizes prediction of $\mathbf{x}^t$ (i.e., $\hat{\mathbf{y}}^t$) for $q(\mathbf{x}^t)$, and uniform weighting assigns equal weights to all weight prototypes. We choose hard weighting for $q(\mathbf{x}^t)$. We will delve into the ablation study for this strategy in Section 4. Finally, $\mathcal{I}_{\mathrm{add.loss}}(\mathbf{x}^t)$ becomes as follows:

$$\mathcal{I}_{\mathrm{add.loss}}(\mathbf{x}^t) \approx \mathcal{I}_{\mathrm{add.loss}}(\mathbf{x}^t, \mathbf{w}_{\tilde{y}^t}) \approx -\cos\_\mathrm{sim}(\nabla_{\boldsymbol{w}}\mathcal{L}(\mathbf{w}_{\tilde{y}^t}; \boldsymbol{w}), \nabla_{\boldsymbol{w}}\mathcal{L}(f_{\boldsymbol{\phi}}(\mathbf{x}^t); \boldsymbol{w})). \tag{9}$$

It is worth noting that , because the last layer is fixed during adaptation, we can compute and store $\nabla_{\boldsymbol{w}}\mathcal{L}(\mathbf{w}; \boldsymbol{w})$ only once before adaptation, which significantly reduces the computational cost.

## 3.4  PROTOTYPICAL INFLUENCE FUNCTION

In Section 3.3, we reformulate the influence function to compute it efficiently in the TTA scenario. Building upon this, we propose the PIF (Prototype Influence Function) regularizer that utilizes the reformulated influence function to increase positive impact of $\mathbf{x}^t$ on the model-trusting data. As mentioned in Section 3.2, $\mathcal{I}_{\mathrm{add.loss}}(\mathbf{x}^t)$ should be negative. The final PIF regularizer for test data $\mathbf{x}^t$ can be expressed as follows:

$$\mathcal{L}_{\mathrm{PIF}}(\mathbf{x}^t) = -\cos\_\mathrm{sim}(\nabla_{\boldsymbol{w}}\mathcal{L}(\mathbf{w}_{\tilde{y}^t}; \boldsymbol{w}), \nabla_{\boldsymbol{w}}\mathcal{L}(f_{\boldsymbol{\phi}}(\mathbf{x}^t); \boldsymbol{w})). \tag{10}$$

Finally, the model updates the parameters of the BN layers by following objectives

$$\mathcal{L}_{\mathrm{total}} = \mathcal{L}_{\mathrm{TTA}} + \alpha\mathcal{L}_{\mathrm{PIF}}. \tag{11}$$

Here, $\mathcal{L}_{\mathrm{TTA}}$ refers to *any* TTA loss function, and $\alpha$ is a hyper-parameter that determines the weight of the PIF regularizer. We empirically discovered the effectiveness of employing a decay technique for the hyper-parameter $\alpha$. Specifically, we set $\alpha$ as $\alpha = \gamma \times (1 + 10 * \frac{iter}{max_{iter}})^{-\beta}$ according to previous study by Yang et al. (2022), where the decay factor $\beta$ regulates the rate of decay and $\gamma$ is default value of $\alpha$.

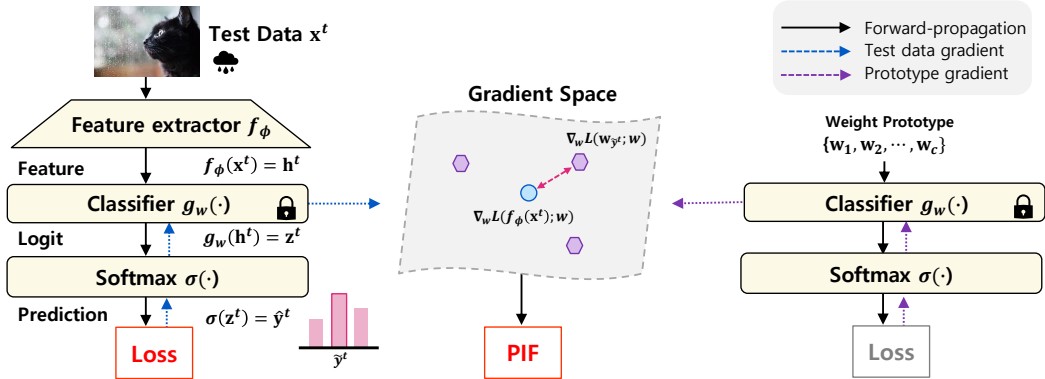

Figure 2: Overview of Prototypical Influence Function for TTA. First, we compute and store the gradients of the last layer for the weight prototypes. Next, we conduct inference and determine the pseudo-label for the test data. Finally, we calculate the PIF that increases the cosine similarity between the gradient of the weight prototype corresponding to the pseudo-label and the last layer gradient of the test data.

In our study, we derive the PIF loss for two common loss functions used in TTA, which are entropy minimization loss and cross-entropy loss. The advantage of these two loss functions is that the gradient of the last layer can be computed efficiently through forward pass alone. Additionally, the PIF approach can be extended to any loss function for which the gradient of the last layer can be computed efficiently. Since the parameters of the last layer include the bias and weights, the gradient of the last layer can be decomposed into gradients for the bias and the weight parameters.

$$\nabla_{\boldsymbol{w}}\mathcal{L}(f_\phi(\mathbf{x}^t); \boldsymbol{w}) = \left[\frac{\partial \mathcal{L}(f_\phi(\mathbf{x}^t); \boldsymbol{w})}{\partial \mathbf{b}}, \frac{\partial \mathcal{L}(f_\phi(\mathbf{x}^t); \boldsymbol{w})}{\partial \mathbf{w}_1}, \cdots, \frac{\partial \mathcal{L}(f_\phi(\mathbf{x}^t); \boldsymbol{w})}{\partial \mathbf{w}_c}\right]. \tag{12}$$

**PIF for entropy minimization loss** When it comes to the entropy minimization loss for test data $\mathbf{x}^t$, it can be defined as $\mathcal{L}_{EM}(f(\mathbf{x}^t); \boldsymbol{w}) = -\sum \hat{\mathbf{y}}^t \log \hat{\mathbf{y}}^t$. In this case, the gradient of $\mathbf{x}^t$ with respect to the last layer can be computed as follows:

$$\frac{\partial \mathcal{L}_{EM}(f(\mathbf{x}^t); \boldsymbol{w})}{\partial \mathbf{b}} = \left(-\frac{\mathbf{z}^t}{S} - C\right), \qquad \frac{\partial \mathcal{L}_{EM}(f(\mathbf{x}^t); \boldsymbol{w})}{\partial \mathbf{w}_i} = \mathbf{h}^t \cdot \left(-\frac{\mathbf{z}^t(i)}{S} - C\right), \tag{13}$$

where $C = \mathbf{z}^t \cdot \exp(\mathbf{z}^t)$ and $S = \sum_{i=1}^c \exp(\mathbf{z}^t)(i)$.

**PIF for cross-entropy loss** The pseudo label of $\mathbf{x}^t$ is $\tilde{y}^t$. Let's represent the one-hot representation of $\tilde{y}^t$ as $\tilde{\mathbf{y}}^t$. Then the cross-entropy loss for test data $\mathbf{x}^t$ can be expressed as $\mathcal{L}_{CE}(\mathbf{x}^t) = -\sum \tilde{\mathbf{y}}^t \log \hat{\mathbf{y}}^t$. In this case, the gradient of $\mathbf{x}^t$ with respect to the last layer can be computed as follows:

$$\frac{\partial \mathcal{L}_{CE}(f(\mathbf{x}^t); \boldsymbol{w})}{\partial \mathbf{b}} = \hat{\mathbf{y}}^t - \tilde{\mathbf{y}}^t, \qquad \frac{\partial \mathcal{L}_{CE}(f(\mathbf{x}^t); \boldsymbol{w})}{\partial \mathbf{w}_i} = \mathbf{h}^t \cdot (\hat{\mathbf{y}}^t - \tilde{\mathbf{y}}^t)(i). \tag{14}$$

When calculating the last layer gradient with respect to the weight prototypes, we can use $\mathbf{w}_i$ as a feature instead of $\mathbf{h}^t$, and apply the same method to obtain it.

**Overall Procedure of PIF** Before conducting the adaptation, we first store the last layer gradients of the weight prototypes. When a batch of test data comes, we compute the statistics of BN layer and perform inference. Then, we update the parameters of BN layer using Equation 11. The detailed pseudo-code is summarized in Algorithm 1 and overview of PIF is illustrated in Figure 2.

---

**Algorithm 1** Prototypical Influence Function for Test-Time Adaptation

---

**Result:** Test-time adapted model
**Input**: Pre-trained model, test data $\mathcal{D}_{\text{te}} = \{\mathbf{x}_j^t\}_{j=1}^{N_{\text{test}}}$
Calculate and save last layer gradient of weight prototype $\{\nabla_{\boldsymbol{w}}\mathcal{L}(\mathbf{w}_1; \boldsymbol{w}), \cdots \nabla_{\boldsymbol{w}}\mathcal{L}(\mathbf{w}_c; \boldsymbol{w})\}$.
**for** *a batch* $\mathcal{X} = \{\mathbf{x}_b^t\}_{b=1}^B$ *in* $\mathcal{D}_{\text{te}}$ **do**
  | Update batch norm statistics using $\mathcal{X}$
  | Perform inference by calculating $\{\hat{\mathbf{y}}_b^t\}_{b=1}^B$
  | Calculate $\mathcal{L}_{\text{PIF}}(\mathcal{X})$ using Equation 10
  | Update parameters of BN layer using Equation 11
**end**

---

## 4 EXPERIMENTS

### 4.1 SETUP

**Datasets and Architectures**    We conduct experiments on two common TTA benchmark datasets: CIFAR-10-C and ImageNet-C. Each dataset has 15 types of corruptions and 5 severity levels. Additionally, we execute experiments with a larger-scale dataset called ImageNet 3D Common Corruptions (ImageNet-3DCC), introduced by Kar et al. (2022), which incorporates the scene's geometry into the transformations, resulting in more realistic corruptions. ImageNet-3DCC consists of 12 distinct types of corruptions, each with five severity levels. For all datasets, we conduct experiments at level 5, which represents the most severe level of corruption. We use Wide-ResNet-28-10 (Zagoruyko & Komodakis, 2016) for CIFAR-10-C data and ResNet-50 with BN layer (He et al., 2016) for both ImageNet-C, ImageNet-3DCC data. We use pretrained models that were trained sufficiently on each clean dataset, namely CIFAR-10 (Krizhevsky et al., 2009) and ImageNet (Russakovsky et al., 2015).

**Baseline Methods**    In our experiments, we evaluate the performance of the PIF regularizer using the following baseline methods.

- **Source** evaluates directly with the pretrained model without adapting to the test data.

- **NORM** (Schneider et al., 2020) updates the BN statistics on the mini-batch samples during test time.

- **TENT** (Wang et al., 2021) updates the statistics and parameters of BN layer by minimizing the entropy minimization loss.

  **EATA** (Niu et al., 2022) employs the Fisher regularizer to safeguard crucial parameter stability and conducts instance selection and re-weighting as part of entropy minimization loss.

  **SAR** (Niu et al., 2023) removes partial noisy samples with large gradients and use reliable entropy minimization methods.

- **PL** (Lee et al., 2013) makes a one-hot pseudo-label by prediction of the model and uses cross-entropy loss to adapt BN parameters to test data.

**Implementation Details**    For the CIFAR-10-C, we employ the pretrained model weights derived from the official implementations of TENT, adhering to the RobustBench protocol (Croce et al., 2020). We set the batch size to 128 and follow the implementations used in TENT, utilizing an NVIDIA GeForce RTX 3090 Ti GPU. For the ImageNet-C and ImageNet-3DCC dataset, we reference the base code from SAR (Niu et al., 2023) and EATA (Niu et al., 2022), following to the implementation details provided in each paper. We set the batch size to 64 and employ an NVIDIA A40 GPU. We report the average performance based on three different random seeds for all experiments.

PIF regularizer incorporates two hyper-parameters: $\gamma$ and $\beta$. The parameter $\beta$ is responsible for modulating the decay rate of $\alpha$, while $\gamma$ represents the default value of $\alpha$. Specifically, we set $\beta = 0$ for the CIFAR-10-C dataset and $\beta = 2$ for the ImageNet-C and ImageNet-3DCC datasets. Detailed values for $\gamma$ are provided in the Appendix.

Table 1: **Classification Error (%)** for each corruption in **CIFAR-10-C** at the highest severity (Level 5). We use **WRN-28-10**. Smaller error is shown in bold.

| Method | Gauss. | Shot | Impul. | Defoc. | Glass | Motion | Zoom | Snow | Frost | Fog | Brit. | Contr. | Elastic | Pixel | JPEG | AVG ↓ |
|---|---|---|---|---|---|---|---|---|---|---|---|---|---|---|---|---|
| Source | 72.3 | 65.7 | 72.9 | 46.9 | 54.3 | 34.8 | 42.0 | 25.1 | 41.3 | 26.0 | 9.3 | 46.7 | 26.6 | 58.5 | 30.3 | 43.5 |
| Norm | 28.5 | 26.3 | 36.1 | 12.9 | 35.2 | 13.9 | 12.2 | 17.5 | 17.8 | 15.2 | 8.4 | 13.3 | 23.6 | 19.9 | 27.7 | 20.6 |
| TENT | 24.8 | 22.4 | 32.0 | 12.1 | 31.8 | 13.4 | 11.0 | 16.1 | 16.4 | 13.8 | 8.2 | 11.6 | 22.0 | 17.1 | 24.3 | 18.5 |
| +PIF(EM) | $22.5_{\pm0.1}$ | $19.9_{\pm0.1}$ | $28.7_{\pm0.3}$ | $11.2_{\pm0.1}$ | $29.2_{\pm0.4}$ | $12.3_{\pm0.2}$ | $10.4_{\pm0.1}$ | $14.3_{\pm0.3}$ | $14.8_{\pm0.1}$ | $12.4_{\pm0.3}$ | $7.8_{\pm0.1}$ | $10.4_{\pm0.3}$ | $20.8_{\pm0.2}$ | $15.3_{\pm0.1}$ | $21.3_{\pm0.2}$ | $16.8_{\pm0.1}$ (−1.7) |
| EATA | 24.6 | 21.9 | 31.7 | 12.1 | 31.0 | 13.1 | 10.8 | 15.6 | 16.3 | 13.3 | 8.1 | 11.2 | 21.6 | 16.6 | 24.0 | 18.1 |
| +PIF(EM) | $22.1_{\pm0.1}$ | $19.4_{\pm0.1}$ | $28.4_{\pm0.3}$ | $10.9_{\pm0.1}$ | $28.8_{\pm0.1}$ | $12.0_{\pm0.0}$ | $10.2_{\pm0.0}$ | $14.0_{\pm0.3}$ | $14.5_{\pm0.1}$ | $12.1_{\pm0.1}$ | $7.8_{\pm0.1}$ | $10.0_{\pm0.2}$ | $20.3_{\pm0.1}$ | $14.9_{\pm0.1}$ | $21.0_{\pm0.3}$ | $16.4_{\pm0.0}$ (−1.7) |
| PL | 26.6 | 25.3 | 34.2 | 12.7 | 33.3 | 14.1 | 12.1 | 16.9 | 17.5 | 14.8 | 8.5 | 12.2 | 23.2 | 18.8 | 26.1 | 19.8 |
| +PIF(CE) | $22.8_{\pm0.2}$ | $20.1_{\pm0.3}$ | $29.3_{\pm0.1}$ | $11.2_{\pm0.1}$ | $29.5_{\pm0.4}$ | $12.4_{\pm0.2}$ | $10.5_{\pm0.1}$ | $14.3_{\pm0.3}$ | $14.9_{\pm0.2}$ | $12.3_{\pm0.3}$ | $7.7_{\pm0.1}$ | $10.4_{\pm0.3}$ | $20.9_{\pm0.1}$ | $15.6_{\pm0.1}$ | $21.8_{\pm0.3}$ | $16.9_{\pm0.1}$ (−2.9) |

Table 2: **Classification Error (%)** for each corruption in **ImageNet-C** at the highest severity (Level 5). We use **RN-50(BN)**. Smaller error is shown in bold.

| Method | Gauss. | Shot | Impul. | Defoc. | Glass | Motion | Zoom | Snow | Frost | Fog | Brit. | Contr. | Elastic | Pixel | JPEG | AVG ↓ |
|---|---|---|---|---|---|---|---|---|---|---|---|---|---|---|---|---|
| Source | 97.8 | 97.1 | 98.2 | 82.1 | 90.2 | 85.2 | 77.5 | 83.1 | 76.7 | 75.6 | 41.1 | 94.6 | 83.1 | 79.4 | 68.3 | 82.0 |
| Norm | 84.8 | 84.2 | 84.2 | 85.0 | 84.6 | 73.7 | 61.1 | 65.7 | 67.0 | 52.0 | 34.8 | 83.1 | 55.9 | 51.0 | 60.2 | 68.5 |
| TENT | 71.4 | 69.4 | 70.0 | 72.0 | 72.9 | 58.7 | 50.8 | 52.8 | 59.1 | 42.4 | 32.6 | 73.9 | 45.3 | 41.5 | 47.8 | 57.4 |
| +PIF(EM) | $69.4_{\pm0.4}$ | $68.1_{\pm0.1}$ | $67.6_{\pm0.1}$ | $69.2_{\pm0.2}$ | $70.6_{\pm0.2}$ | $55.3_{\pm0.1}$ | $49.3_{\pm0.1}$ | $50.3_{\pm0.1}$ | $57.4_{\pm0.3}$ | $41.3_{\pm0.1}$ | $32.6_{\pm0.2}$ | $68.1_{\pm1.5}$ | $43.6_{\pm0.1}$ | $40.2_{\pm0.0}$ | $46.2_{\pm0.1}$ | $55.3_{\pm0.1}$ (−2.1) |
| EATA | 65.2 | 63.0 | 66.5 | 66.8 |  | 53.2 | 47.2 | 48.4 | 54.3 | 40.0 | 31.9 | 55.5 | 42.1 | 39.5 | 44.9 | 52.2 |
| +PIF(EM) | $64.5_{\pm0.2}$ | $62.2_{\pm0.2}$ | $63.2_{\pm0.1}$ | $65.8_{\pm0.6}$ | $65.8_{\pm0.1}$ | $51.4_{\pm0.2}$ | $46.7_{\pm0.1}$ | $47.1_{\pm0.1}$ | $53.4_{\pm0.2}$ | $39.4_{\pm0.1}$ | $32.1_{\pm0.1}$ | $53.8_{\pm0.4}$ | $41.4_{\pm0.1}$ | $39.1_{\pm0.2}$ | $44.4_{\pm0.1}$ | $51.3_{\pm0.1}$ (−0.9) |
| SAR | 69.7 | 69.6 | 69.0 | 71.5 | 71.5 | 58.1 | 50.7 | 52.9 | 57.9 | 42.4 | 32.7 | 63.2 | 45.5 | 41.6 | 47.7 | 56.3 |
| +PIF(EM) | $69.2_{\pm0.2}$ | $68.9_{\pm0.5}$ | $68.1_{\pm0.2}$ | $70.9_{\pm0.1}$ | $70.9_{\pm0.4}$ | $57.5_{\pm0.2}$ | $50.4_{\pm0.2}$ | $52.4_{\pm0.1}$ | $57.5_{\pm0.1}$ | $42.2_{\pm0.2}$ | $32.6_{\pm0.0}$ | $61.2_{\pm0.7}$ | $45.1_{\pm0.0}$ | $41.4_{\pm0.1}$ | $47.4_{\pm0.1}$ | $55.7_{\pm0.1}$ (−0.6) |
| PL | 74.2 | 72.5 | 73.1 | 74.8 | 75.5 | 62.2 | 52.8 | 55.7 | 60.7 | 44.2 | 33.1 | 75.2 | 47.7 | 43.3 | 50.1 | 59.7 |
| +PIF(CE) | $71.3_{\pm0.5}$ | $70.1_{\pm0.8}$ | $69.3_{\pm0.1}$ | $71.1_{\pm0.5}$ | $71.7_{\pm0.3}$ | $56.9_{\pm0.4}$ | $50.1_{\pm0.1}$ | $51.4_{\pm0.2}$ | $57.6_{\pm0.2}$ | $42.0_{\pm0.1}$ | $32.9_{\pm0.1}$ | $66.6_{\pm3.4}$ | $44.4_{\pm0.1}$ | $40.9_{\pm0.1}$ | $46.8_{\pm0.3}$ | $56.2_{\pm0.3}$ (−3.5) |

Table 3: **Classification Error (%)** for each corruption in **ImageNet-3DCC** at the highest severity (Level 5). We use **RN-50(BN)**. Smaller error is shown in bold.

| Method | Near_focus | Far_focus | Fog_3d | Flash | Color_quant. | Low_light | XY_motion. | Z_motion. | ISO_noise | Bit_error | H265_ABR | H265_CR | AVG ↓ |
|---|---|---|---|---|---|---|---|---|---|---|---|---|---|
| Source | 99.9 | 99.9 | 99.9 | 100.0 | 99.9 | 99.9 | 99.9 | 99.9 | 99.9 | 100.0 | 99.9 | 99.9 | 99.9 |
| Norm | 45.4 | 55.0 | 75.1 | 80.9 | 71.8 | 64.1 | 79.1 | 67.4 | 76.9 | 91.8 | 80.8 | 76.9 | 72.1 |
| TENT | 40.1 | 49.6 | 68.5 | 75.6 | 62.4 | 50.6 | 70.5 | 57.2 | 62.1 | 91.6 | 75.7 | 70.3 | 64.5 |
| +PIF(EM) | $39.4_{\pm0.1}$ | $48.9_{\pm0.2}$ | $68.2_{\pm0.3}$ | $75.0_{\pm0.0}$ | $61.3_{\pm0.1}$ | $49.1_{\pm0.0}$ | $69.0_{\pm0.0}$ | $55.9_{\pm0.0}$ | $60.6_{\pm0.2}$ | $92.2_{\pm0.1}$ | $75.7_{\pm0.0}$ | $69.9_{\pm0.2}$ | $63.8_{\pm0.0}$ (−0.7) |
| EATA | 38.5 | 47.7 | 62.6 | 71.3 | 59.2 | 47.4 | 65.1 | 53.0 | 57.3 | 89.6 | 71.7 | 66.5 | 60.8 |
| +PIF(EM) | $38.3_{\pm0.2}$ | $47.3_{\pm0.0}$ | $61.7_{\pm0.3}$ | $70.3_{\pm0.1}$ | $58.5_{\pm0.0}$ | $46.7_{\pm0.1}$ | $64.0_{\pm0.1}$ | $52.3_{\pm0.2}$ | $56.5_{\pm0.1}$ | $89.9_{\pm0.3}$ | $71.3_{\pm0.1}$ | $66.2_{\pm0.1}$ | $60.3_{\pm0.0}$ (−0.5) |
| SAR | 40.4 | 50.0 | 65.9 | 73.9 | 62.0 | 50.4 | 69.1 | 56.9 | 61.2 | 89.9 | 73.9 | 68.9 | 63.5 |
| +PIF(EM) | $40.2_{\pm0.1}$ | $49.8_{\pm0.2}$ | $65.4_{\pm0.1}$ | $73.6_{\pm0.0}$ | $61.6_{\pm0.1}$ | $50.1_{\pm0.1}$ | $68.7_{\pm0.1}$ | $56.5_{\pm0.2}$ | $60.7_{\pm0.1}$ | $90.3_{\pm0.5}$ | $73.7_{\pm0.1}$ | $68.6_{\pm0.1}$ | $63.2_{\pm0.0}$ (−0.3) |
| PL | 41.3 | 51.0 | 69.8 | 77.2 | 64.3 | 53.2 | 72.5 | 59.4 | 64.8 | 91.0 | 75.9 | 71.1 | 65.9 |
| +PIF(CE) | $40.0_{\pm0.1}$ | $49.5_{\pm0.1}$ | $69.2_{\pm0.2}$ | $75.6_{\pm0.2}$ | $62.3_{\pm0.1}$ | $50.6_{\pm0.1}$ | $70.8_{\pm0.2}$ | $57.4_{\pm0.1}$ | $62.4_{\pm0.3}$ | $91.2_{\pm0.1}$ | $75.2_{\pm0.2}$ | $70.1_{\pm0.0}$ | $64.5_{\pm0.0}$ (−1.4) |

## 4.2 EXPERIMENTAL RESULTS

We assess the efficacy of the proposed PIF loss by evaluating the classification error after incorporating the PIF regularizer into the baseline methods. The results obtained from three benchmark datasets are shown in Tables 1, 2 and 3 with performance improvements highlighted in red.

**Results on CIFAR-10-C and ImageNet-C** Table 1 displays the CIFAR-10-C dataset results, while Table 2 showcases the results for the ImageNet-C dataset. Incorporating the PIF regularizer into the baseline methods consistently leads to a reduction in classification errors across various corruption types for both datasets when compared to the baseline approaches. Remarkably, when compared to the PL, the utilization of the PIF regularizer results in a significant performance improvement of 2.9% for the CIFAR-10-C dataset and 3.5% for the ImageNet-C dataset.

**Results on ImageNet-3DCC** Table 3 shows the results for the ImageNet-3DCC dataset. Similar to the smaller datasets, the PIF loss consistently enhance performance in terms of overall average performance, even though this dataset is highly realistic and challenging to adapt to. It demonstrates that incorporating the PIF loss consistently aids in better adaptation without impeding the original learning process.

## 4.3 ABLATION STUDY

**Effects of IF approximation strategies**  We present the performance comparison between cosine similarity and dot product as an approximation for computing the influence function in Figure 3. The results clearly demonstrate the superiority of cosine similarity. By using cosine similarity as an approximation for the influence function, the scale of gradients is ignored, which can enhance learning stability and reduce errors caused by random projection, approximating the Hessian matrix to the identity matrix.

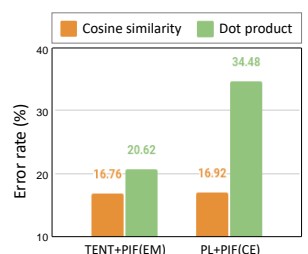

Figure 3: **Average classification Error (%) on CIFAR-10-C**

**Effects of weighting strategies**  In Equation 8, $q(\mathbf{x}^t)$ determines the extent of weighting applied to each weight prototype. There are three options: hard weighting, soft weighting, and uniform weighting. Table 4 presents the outcomes of these weighting strategies. The experimental results reveal a significant performance deterioration in $\text{PIF}_{\text{unif}}$ when combined with TENT on the CIFAR-10-C dataset, compared to the other variants. This decline can be attributed to the lower reliability of the IF for classes

Table 4: **Average classification Error (%)** for each weighting strategy. Smaller error is shown in bold.

| Dataset | Loss | hard | soft | uniform |
|---|---|---|---|---|
| CIFAR-10-C | TENT+PIF(EM) | **16.76** | 16.83 | 34.85 |
| | PL+PIF(CE) | **16.92** | 18.00 | 17.59 |
| ImageNet-C | TENT+PIF(EM) | **55.33** | 57.37 | 57.38 |
| | PL+PIF(CE) | **56.20** | 59.64 | 59.65 |
| ImageNet-3DCC | Tent+PIF(EM) | **63.77** | 64.51 | 64.51 |
| | PL+PIF(CE) | **64.51** | 65.97 | 65.95 |

with high predicted probabilities. Additionally, $\text{PIF}_{\text{soft}}$ exhibits lower performance and higher computational costs compared to $\text{PIF}_{\text{hard}}$ because $\text{PIF}_{\text{soft}}$ utilizes the IF for all classes, unlike $\text{PIF}_{\text{hard}}$. Given that $\text{PIF}_{\text{hard}}$ consistently demonstrates strong performance across all cases, we have chosen the hard weighting approach.

**Effects of hyper-parameters**  TTA, the online adaptation to test data distribution, necessitates careful consideration of hyper-parameter sensitivity, as it is a critical factor related to the applicability of the model. Numerous or sensitive hyper-parameters may require inefficient additional tuning, which violates the online property of TTA. The PIF regularizer involves two hyperparameter: $\gamma$ and $\beta$. Key hyper-parameter is $\gamma$ which represents the default value of $\alpha$. Figure 4 showcases the average errors as the hyper-parameters $\gamma$ vary while adapting to the CIFAR-10-C dataset. It illustrates the performance enhancements achieved by incorporating PIF compared to when it is not used. Additionally, the figure highlights the sustained robust performance of the hyperparameter beyond a certain threshold. In essence, these findings suggest the possibility of applying various TTA methods with minimal tuning, underscoring the simplicity and effectiveness of this approach.

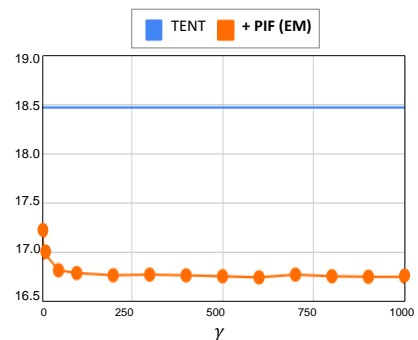

Figure 4: Sensitivity of the average classification error on CIFAR-10-C with respect to the hyperparameter $\gamma$.

## 5  CONCLUSION

In this study, we introduce a novel method called the Prototypical Influence Function (PIF) to enhance the performance of test-time adaptation (TTA) scenarios. We aim to adapt the model to test data without increasing the loss of the model-trusting data, ensuring that the model generalizes well to unseen data. To achieve this, we quantify the impact of test data on the loss of model-trusting data by utilizing influence function. We reformulated the influence function to address computational bottlenecks by approximating the Hessian as an identity matrix and computing gradients only for the last layer. Additionally, due to our limited access to the entire test dataset, we approximate the model-trusting data using the weight prototype obtained from the last layer of the model. Our PIF method optimizes the model by maximizing the positive influence of the test data on the weight prototype. The resulting PIF regularizer is added to the existing loss during TTA. The PIF regularizer is applicable to any loss function for which the last layer gradient can be computed efficiently. We experimentally show the effectiveness and robustness of the PIF regularizer.

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

## A    PROOFS OF INFLUENCE OF TEST DATA ON THE MODEL

This derivation is based on (Koh & Liang, 2017). Remember that the estimated parameter $\hat{\boldsymbol{\theta}}$ minimizes the empirical risk.

$$\hat{\boldsymbol{\theta}} \stackrel{\text{def}}{=} \operatorname*{argmin}_{\boldsymbol{\theta} \in \boldsymbol{\Theta}} \mathcal{L}(\mathcal{D}_{tr}; \boldsymbol{\theta}). \tag{15}$$

Additionally, we make the assumption that $\mathcal{L}(\mathcal{D}_{tr}; \boldsymbol{\theta})$ is twice-differentiable and strongly convex with respect to $\boldsymbol{\theta}$. In other words, the Hessian matrix $H_{\hat{\boldsymbol{\theta}}}$, defined as $\nabla_{\boldsymbol{\theta}}^2 \mathcal{L}(\mathcal{D}_{tr}; \hat{\boldsymbol{\theta}})$ in Equation 16, exists and is positive definite.

This ensures the existence of the inverse of $H_{\hat{\boldsymbol{\theta}}}$, denoted as $H_{\hat{\boldsymbol{\theta}}}^{-1}$, which will be utilized in the subsequent derivation.

$$H_{\hat{\boldsymbol{\theta}}} \stackrel{\text{def}}{=} \nabla_{\boldsymbol{\theta}}^2 \mathcal{L}(\mathcal{D}_{tr}; \hat{\boldsymbol{\theta}}). \tag{16}$$

The perturbed parameters $\hat{\boldsymbol{\theta}}_{\epsilon, \mathbf{x}^t}$ can be written as

$$\hat{\boldsymbol{\theta}}_{\epsilon, \mathbf{x}^t} \stackrel{\text{def}}{=} \operatorname*{argmin}_{\boldsymbol{\theta} \in \boldsymbol{\Theta}} \mathcal{L}(\mathcal{D}_{tr}; \boldsymbol{\theta}) + \epsilon \mathcal{L}(\mathbf{x}^t; \boldsymbol{\theta}) \tag{17}$$

Let's define the parameter change as $\Delta_\epsilon = \hat{\boldsymbol{\theta}}_{\epsilon, \mathbf{x}^t} - \hat{\boldsymbol{\theta}}$. It's worth noting that since $\hat{\boldsymbol{\theta}}$ does not depend on $\epsilon$, the quantity we aim to calculate can be expressed in terms of it:

$$\frac{d\hat{\boldsymbol{\theta}}_{\epsilon, \mathbf{x}^t}}{d\epsilon} = \frac{d\Delta_\epsilon}{d\epsilon} \tag{18}$$

Given that $\hat{\boldsymbol{\theta}}_{\epsilon, \mathbf{x}^t}$ minimizes Equation 17, let's analyze its first-order optimality conditions:

$$0 = \nabla \mathcal{L}(\mathcal{D}_{tr}; \hat{\boldsymbol{\theta}}_{\epsilon, \mathbf{x}^t}) + \epsilon \nabla \mathcal{L}(\mathbf{x}^t; \hat{\boldsymbol{\theta}}_{\epsilon, \mathbf{x}^t}) \tag{19}$$

Subsequently, as $\hat{\boldsymbol{\theta}}_{\epsilon, \mathbf{x}^t} \to \hat{\boldsymbol{\theta}}$ when $\epsilon \to 0$, we can conduct a Taylor expansion of the right-hand side:

$$0 \approx \left[ \nabla \mathcal{L}(\mathcal{D}_{tr}; \hat{\boldsymbol{\theta}}) + \epsilon \mathcal{L}(\mathbf{x}^t; \hat{\boldsymbol{\theta}}) \right] + \left[ \nabla^2 \mathcal{L}(\mathcal{D}_{tr}; \hat{\boldsymbol{\theta}}) + \epsilon \nabla^2 \mathcal{L}(\mathbf{x}^t; \hat{\boldsymbol{\theta}}) \right] \Delta_\epsilon \tag{20}$$

where we have dropped $o(\|\Delta_\epsilon\|)$ terms.

By solving for $\Delta_\epsilon$, we obtain:

$$\Delta_\epsilon \approx - \left[ \nabla^2 \mathcal{L}(\mathcal{D}_{tr}; \hat{\boldsymbol{\theta}}) + \epsilon \nabla^2 \mathcal{L}(\mathbf{x}^t; \hat{\boldsymbol{\theta}}) \right]^{-1} \left[ \nabla \mathcal{L}(\mathcal{D}_{tr}; \hat{\boldsymbol{\theta}}) + \epsilon \nabla \mathcal{L}(\mathbf{x}^t; \hat{\boldsymbol{\theta}}) \right] \tag{21}$$

Since $\hat{\boldsymbol{\theta}}$ minimizes $\mathcal{L}(\mathcal{D}_{tr}; \boldsymbol{\theta})$, $\nabla \mathcal{L}(\mathcal{D}_{tr}; \hat{\boldsymbol{\theta}}) = 0$. Dropping $o(\epsilon)$ terms, we have

$$\Delta_\epsilon \approx -\nabla^2 \mathcal{L}(\mathcal{D}_{tr}; \hat{\boldsymbol{\theta}})^{-1} \nabla \mathcal{L}(\mathbf{x}^t; \hat{\boldsymbol{\theta}}) \times \epsilon \tag{22}$$

By combining Equation 16 and 18, we can deduce that:

$$\left. \frac{d\hat{\boldsymbol{\theta}}_{\epsilon, \mathbf{x}^t}}{d\epsilon} \right|_{\epsilon=0} = -H_{\hat{\boldsymbol{\theta}}}^{-1} \nabla_{\boldsymbol{\theta}} \mathcal{L}(\mathbf{x}^t; \hat{\boldsymbol{\theta}}) \stackrel{\text{def}}{=} \mathcal{I}_{\text{add.params}}(\mathbf{x}^t) \tag{23}$$

## B  IMPLEMENTATION DETAILS

The hyperparameter $\gamma$ which is responsible for the default value of $\alpha$ is presented in Table 5 for each dataset and baseline method. Code will be included in the final version of our git repo.

Table 5: Hyperparameter $\gamma$

| Method | CIFAR-10-C | ImageNet-C | ImageNet-3DCC |
|---|---|---|---|
| TENT+PIF(EM) | 500 | 150 | 90 |
| EATA+PIF(EM) | 200 | 55 | 30 |
| SAR+PIF(EM) | $\times$ | 2900 | 2500 |
| PL+PIF(CE) | 500 | 200 | 90 |

