# OpenReview forum: "Prototypical Influence Function for Fully Test-time Adaptation"
_ICLR.cc/2024/Conference — Submitted to ICLR 2024_

### Official Review · Reviewer_yBgF · 2023-10-23

**Soundness:** 3 good
**Presentation:** 4 excellent
**Contribution:** 3 good
**Rating:** 5
**Confidence:** 4

**Summary:**

This paper studied the problem of test-time adaptation which directly performed model adaptation and inference on test data. To solve this problem, this paper introduced a novel prototypical influence function (PIF) to regularize the adaptation process. Furthermore, in order to handle the high computational complexity of the original PIF regularizer, this paper presented two efficient approximations with respect to the influence function and the class prototype. Experimental results supported the superior performance of the proposed PIF regularizer on multiple benchmark data sets.

**Strengths:**

**Originality:** This paper studied the test-time adaptation by investigating the influence of test data on the loss of model-trusting data. This motivated the development of a prototypical influence function (PIF) regularizer. The computationally expensive hessian matrix and unavailable model-trusting data might limit the applications of PIF. Thus, this paper further provided more practical approximations of PIF and validated the effectiveness of these approximations on several benchmarks.

**Quality:** The proposed PIF regularizer is well-motivated. It is formulated by the influence of test data on the model parameters. The proposed approximations also allow the PIF regularizer to be efficiently computed in real-world test-time adaptation scenarios. The experiments also demonstrate that with the proposed PIF regularizer, existing test-time adaptation methods can achieve much better performance.

**Clarity:** The paper is well-written. It clearly derived the PIF regularizer as well as its approximations. The experimental settings are easy to follow. The ablation studies on the approximation strategies and the weighting strategies further validate the superior training procedures given in Alg. 1.

 **Significance:** Test-time adaptation is a practical solution for efficient inference of a pre-trained model on the corrupted test data. The derived influence function in this paper provides a solid explanation on the adaptation process during testing. The fast approximation further allows the proposed techniques to be applied to real-world scenarios.

**Weaknesses:**

(1) The major concern of the PIF regularizer is the approximation error over the influence function and the class prototype.

- First, it approximates the Hessian matrix to the identity matrix and then uses the cosine similarity between two gradients. Though empirical evaluation shows the good performance of such approximations, it is unclear regarding the approximation error between this approximation and the original term in Eq. (4), and how this error affects the adaptation performance.
- Second, it approximates the model-trusting data with the weight prototypes. It is confusing whether the weight prototypes can always be positively associated with the unseen model-trusting data during testing. Considering the distribution shift between training and test data, it might be more convincing to explain under what conditions, the correlation between the weight prototypes and the unseen model-trusting data can guarantee the accurate approximation of Eq. (4).

(2) The computational complexity of Algorithm 1 is not analyzed, especially for the step of updating parameters of the BN layer using Eq. (11). More specifically, what is the complexity of the PIF regularizer with respect to the parameters of BN layers?

(3) The selection of hyperparameters is unclear. The experiments show that different values of $\gamma$ and $\beta$ are used in different data sets. However, it is not explained how these hyperparameters are selected in the test-time adaptation settings.

**Questions:**

(1) Why is the weighting strategy applied for deriving the approximation in Eq. (8)?

(2) It might be more convincing to provide the impact of PIF on the test-time adaptation regarding the adaptation efficiency.




################################################################

Since the authors did not address my concerns, I would like to revise my score from 6 to 5. The major concerns include (1) the gap between the approximated and exact calculation of IF values; (2) the unclear assumption behind weight prototypes; (3) the unclear hyperparameter optimization; and (4) the unexplained weighting strategy.

---

> ### Author Response · Authors · 2023-11-20
> **Response to Reviewer yBgF**
>
> We thank you for the insightful comments and suggestions. We have addressed each of your questions below.
>
> **W1-1.** It approximates the Hessian matrix to the identity matrix and then uses the cosine similarity between two gradients. Though empirical evaluation shows the good performance of such approximations, it is unclear regarding the approximation error between this approximation and the original term in Eq. (4), and how this error affects the adaptation performance.
>
> **A.** Our objective is not to precisely calculate IF values but rather to maximize them, ensuring that test data positively influences prototype loss. As training progresses, the relative magnitude of IF values becomes crucial, and the exact calculation does not significantly impact adaptation performance.
>
> **W1-2.** It approximates the model-trusting data with the weight prototypes. It is confusing whether the weight prototypes can always be positively associated with the unseen model-trusting data during testing. Considering the distribution shift between training and test data, it might be more convincing to explain under what conditions, the correlation between the weight prototypes and the unseen model-trusting data can guarantee the accurate approximation of Eq. (4).
>
> **A.** We define model-trusting data as virtual data that produces near one-hot predictions. Considering how the last layer operates, for near one-hot predictions, it needs to align well with the last layer weights. Thus, we can consider the last layer weight vector itself as the prototype for model-trusting data. For effective use of model-trusting data, there should not be a significant distribution shift between source and target data, and the model should share the learned embedding space.
>
> **W2.** The computational complexity of Algorithm 1 is not analyzed, especially for the step of updating parameters of the BN layer using Eq. (11). More specifically, what is the complexity of the PIF regularizer with respect to the parameters of BN layers?
>
> **A.** The PIF value can be computed using only forward pass values, and its complexity concerning BN layer parameters is the same as that of EM loss..
>
> **W3.** The selection of hyperparameters is unclear. The experiments show that different values of � and � are used in different data sets. However, it is not explained how these hyperparameters are selected in the test-time adaptation settings.
>
> **A.** We performed a greedy search for optimal hyperparameters.
>
> **Q1.** Why is the weighting strategy applied for deriving the approximation in Eq. (8)?
>
> **A.** The weighting strategy in designing the PIF regularizer is crucial as it determines which class's prototype gradient to align with, given the absence of labels for test data.
>
> **Q2.** It might be more convincing to provide the impact of PIF on the test-time adaptation regarding the adaptation efficiency.
>
> **A.** Since the last layer is fixed, prototype gradients can be precomputed before training. Additionally, the test data gradient is only computed for the last layer, making it computationally efficient using forward pass calculations exclusively.

---

> > ### Comment · Reviewer_yBgF · 2023-11-20
> > **Comments**
> >
> > Thanks for sharing the rebuttals. However, I did have several follow-up questions after reading the rebuttals.
> >
> > (1) It is still confusing why the exact calculation does not significantly impact adaptation performance. Are there any theoretical and empirical results supporting this argument?
> >
> > (2) Given that there should not be a significant distribution shift between source and target data, it might be more convincing to point out under what conditions the proposed method based on weight prototypes can be applied in real-world TTA scenarios.
> >
> > (3) It used the greedy search for optimal hyperparameters. Does it select the best hyperparameters based on test accuracy? Or any validation method is used here?
> >
> > (4) Eq. (8) shows the approximation between $\sum\_{k=1}^M \mathcal{I}\_{add.loss}(x^t, x_k^m)$ and $\sum_{i=1}^c q(x^t)(i) \times \mathcal{I}_{add.loss}(x^t, w_i)$. But it is not explained how this approximation is derived and what is the approximation error. This can also help explain why the weighting strategy is necessary in this case.

---

### Official Review · Reviewer_SDLt · 2023-10-30

**Soundness:** 1 poor
**Presentation:** 2 fair
**Contribution:** 2 fair
**Rating:** 5
**Confidence:** 3

**Summary:**

The paper propose to regularize Test-Time Adaptation methods by utilizing Prototypical Influence Function (PIF) for regularization. It uses the influence function to assess the influence of adapting a test data point on the loss for high-confident samples. To make the calculation of the influence function practical for TTA the authors propose some approximations. The proposed TTA with PIF regularization results the accuracy of some TTA methods.

**Strengths:**

S1. The idea to introduce prototypical functions is interesting (and challenging). The authors proposed two original modifications to the original influence function calculation to make the computation feasible.
S2. Evaluations show the effectiveness of the proposed method.

**Weaknesses:**

W1. There are many details missing about the hyper-parameters:

  a) From the implementation details it looks like the method requires specific parameter fine tuning-per method. As in the Supplement B, for each dataset and each baseline method, different sets of parameters were used. Can we at least use one set of parameters for a dataset?
Fig. 4 suggests that we can simply use large gamma parameter and achieve good results. But in the Supplement B some very small values were used. Can we see Fig. 4 for example ImageNet-C + EATA or ImageNet-3DCC + PIL?
Also for SAR very large gamma value was used (larger than maximum value in Fig. 4). Can You explain here? Also there seems to be a typo in the table for CIFAR-10-C and SAR

  b) a similar question arises for the Beta parameter. Beta = 0 for CIFAR and 2 for ImageNet. What happened if other parameters were used?

  c) I assume oracle hyper-parameter selection was used to compute the optimal parameters for the proposed method. Can the authors provide how the search was run, i.e., what was the range of parameters?

  d) What was the learning rate in the experiments - I assume it was taken from the baseline model. When adding your method, do you use the same learning rate as the baseline or also tune it separately?

W2. How is the model-trusting data defined? I assume you have used some entropy or maximum confidence threshold, but no details are given in the paper. Also, how sensitive is the method to this choice?

W3. The method is evaluated only on methods that update Batch Norms. Can we apply it to some other methods that update the whole model? For example COTTA, AdaContrast, Robust Mean Teacher?

W4. The gamma parameter currently uses decay-technique which requires setting a maximum number of iterations. This is a serious problem for TTA as we cannot know the maximum number of iterations. Can authors elaborate on that? Ideally, the authors would use a technique here that does not require maximum number of iterations.

W5. There is no source-code provided in the supplement which reduces the potential impact. It would be great to provide some version of the code for the rebuttal.

W6. In the contributions the authors mention: "PIF regularizer which satisfies the limited access and low-resource constraints of TTA" - however, the low-resources constraint is never analyzed or discussed in depth later on.

**Questions:**

Q1. Batch size of 128 was used in the experiments. What would happen if we used some smaller batch size, for example 16? Are the gains of the proposed method more visible at large batches?

Q2.  About model-trusting data. It is referred a lot with different context, but it is not clear why is it that important. It is defined as data with low entropy.
In one place the author writes „It is crucial to minimize the loss of model-trusting data in TTA”. This sentence for example requires more explanation, i.e., if the model is already good at predicting those samples, it is not clear that reducing the loss on those samples will further improve the accuracy?
Is the motivation here that we want to focus on optimizing only the samples with high entropy, to focus only on „reliable” samples, as it is commonly used in methods based on pseudo-labeling?

Q3. In Fig. 1, do you need some extra normalization for the models weights?

Q4. How the changes in the feature extractor (even as small as chaning the BN layers parameters) influance the calculation of PIF? How the method works for a longer sequences?

---

> ### Author Response · Authors · 2023-11-20
> **Response to Reviewer SDLt**
>
> We appreciate the detailed questions and comments. Below are our responses:
>
> **W1.** There are many details missing about the hyper-parameters:
>
> **a)** We applied PIF to TENT, EATA, PL, and SAR for each dataset, and each method has different loss functions, making it impossible to use the same hyperparameters across methods.
>
> For ImageNet-C + EATA, an ablation study on the hyperparameter gamma yielded the following results:
>
> | gamma      | 10    | 20    | 30    | 40    | 50    | 60    | 70    | 80    | 90    | 100   |
> |------------|-------|-------|-------|-------|-------|-------|-------|-------|-------|-------|
> | error_rate | 51.74 | 51.51 | 51.38 | 51.35 | 51.37 | 51.43 | 51.51 | 51.62 | 51.81 | 52.03 |
>
> SAR requires a larger gamma due to applying PIF only in the first update during backpropagation.
>
> **b)** Beta values were chosen based on experimentation. For TENT+PIF on ImageNet-C, fixing gamma at 150 and varying beta produced the following error rates:
>
> | beta       | 0    | 0.5  | 1    | 1.5  | 2    | 3    | 4    | 5    | 6    |
> |------------|------|------|------|------|------|------|------|------|------|
> | error rate | 64.8 | 58.4 | 55.9 | 55.3 | 55.3 | 55.6 | 55.9 | 56.2 | 56.3 |
>
> **c)** We performed a greedy search for optimal hyperparameters.
>
> **d)** We use the same learning rate as the baseline.
>
>
> **W2.** How is the model-trusting data defined?
>
> **A.** Model-trusting data refers to samples with low entropy, similar to reliable samples in other methods. Adapting the model with data that gives near one-hot predictions is crucial.
>
>
> **W3.** Can the method be applied to other methods updating the whole model?
>
> **A*.* Yes, but frequent updates to the last layer parameters would reduce memory and time efficiency.
>
>
> **W4.** The gamma parameter currently uses decay-technique which requires setting a maximum number of iterations.
>
> **A.** We used a maximum number of iterations for decay techniques, but for cases like CIFAR-10, good performance was observed without decay. Additionally, for ImageNet-C, TENT+PIF showed a 1.4% improvement without decay, and PL showed a 2.7% improvement without decay.
>
>
> **W5.** There is no source-code provided in the supplement.
>
> **A.** The source code will be made public when the paper is published.
>
>
> **W6.** The low-resources constraint is never analyzed or discussed in depth later on.
>
> **A.** To utilize the influence function directly, heavy computations like Hessian inverse and full model gradient calculations are needed. We propose various approximations to calculate the influence function values through forward propagation only.
>
>
> **Q1.** What would happen if a smaller batch size, for example, 16, was used?
>
> **A.** For TENT+PIF on CIFAR-10-C, the error rate decreased from 22.7% to 21.7% with a batch size of 16, showing a slight performance improvement compared to a batch size of 128. The reduced performance improvement is attributed to increased noise in the gradient with smaller batch sizes.
>
>
> **Q2.** Regarding model-trusting data, if the model is already good at predicting those samples, is it clear that reducing the loss on those samples will further improve accuracy?
>
> **A.** The author of EATA empirically showed that adapting model with reliable sample (test data with low entropy) is critical. To selectively adapt the model with the reliable samples, EATA introduced a threshold hyperparameter and filtered the test data for the adaptation. On the contrary, we used model trusting data, which is similar to the reliable samples but hypothetical. Considering that the reliable samples have low entropy which leads to near one-hot prediction, the observation from EATA can be extended that it is crucial to adapt model with test data which gives near one-hot prediction (model trusting data)
>
>
> **Q3.** Do you need extra normalization for the model's weights in Fig. 1?
>
> **A.** In Fig. 1, the model weights and test data features were normalized for plotting.
>
>
> **Q4.** How do changes in the feature extractor or longer sequences influence PIF calculation?
>
> **A.** Changes in the feature extractor affect the input values of the last layer, influencing the calculation of PIF. PIF aligns the last layer gradient of the prototype with the last layer gradient of the test data, and its effectiveness is independent of sequence length.
>
>
> We hope these responses address your queries and concerns.

---

### Official Review · Reviewer_bVft · 2023-10-31

**Soundness:** 3 good
**Presentation:** 3 good
**Contribution:** 2 fair
**Rating:** 3
**Confidence:** 4

**Summary:**

The paper introduces a TTA method that calculates influence function as supervised regularization signals for adaptations. In order to apply the influence function, the authors make two modifications to adapt the method to the TTA setting. By appending the PIF regularizer to previous TTA methods, the authors conduct experiments to demonstrate the performance improvements.

**Strengths:**

1. The authors claim that they are the first paper to apply influence function (IF) to TTA settings.
2. The authors make efforts to address the difficulties of applying IF to TTA.
3. Diverse experiments are conducted to show the effectiveness of the method.
4. Overall, the paper is well-organized and technically sound.

**Weaknesses:**

1. The presentation of the prototypical approximation is confusing. According to my understanding, due to test data being inaccessible, instead of using model-trusting test data, the authors are actually comparing the target test sample with the model's original parameters or original training data. Therefore, the method considers the model as a prototypical network and aims to align the target test sample's gradient to the prototype's gradients. That means the overall purpose of the method is aligning the gradients of the two loss terms in $\hat{\boldsymbol{\theta}}\_{\epsilon, \mathbf{x}^t} \stackrel{\text{ def }}{=} \operatorname{argmin}\_{\boldsymbol{\theta} \in \Theta} \mathcal{L}\left(\mathcal{D}_{\text {tr }} ; \boldsymbol{\theta}\right)+\epsilon \mathcal{L}\left(\mathbf{x}^t ; \boldsymbol{\theta}\right)$. The only difference is using prototypes to replace $\mathcal{D}\_{tr}$. If I'm right, I don't see any reason to complicate the introduction of this method like what it was in the paper.
2. The experimental performance seems not prominent. And for different weighting strategies, only hard weight works fine. While the improvements are limited, the method introduces two extra hyperparameters, which raises concerns about whether using this PIF regularizer is a practical option.

**Questions:**

1. Why didn't the authors compare SAR with LayerNorm and GroupNorm? As shown in SAR, SAR works much better using LN and GN. If PIF cannot be used in LN and GN and also cannot beat the results, can you convince me why I need PIF? Or can the author figure out how PIF can be used in LN and GN cases?
2. If I understand the paper right, PIF is basically aligning test samples with prototypes in gradient space. In such a case, one natural question is have you tried aligning them in embedding space? How does it perform? If I can align the first-order derivative, can I align the second-order? Will it provide a better performance?

Overall, the paper is interesting, but the contributions are not enough in my opinion.

---

> ### Author Response · Authors · 2023-11-18
> **Response to Reviewer bVft**
>
> Thank you for acknowledging that our paper is well-organized and technically sound. Building on the comments you provided, we were able to further demonstrate the robustness and effectiveness of PIF regularizer by incorporating additional theoretical explanations and experiments. We are truly grateful for the valuable comments.
>
> **W1.** The presentation of the prototypical approximation is confusing.
>
> **A.** Core constraints of TTA are "domain shift in test data", "inference and adaptation in parallel", and "unlabeled online test data".
> Since TTA presumes the domain shift between train and test data, it is not straight-forward aligning two quantities from train and test time (gradient of L(tr) and L(test).) This is why we introduced the concept of influence function to derive the interpretable regularizer.
> The prototypical approximation is derived from observations in EATA, the SOTA method. The author of EATA empirically showed that adapting model with reliable sample (test data with low entropy) is critical. To selectively adapt the model with the reliable samples, eata introduced a threshold hyperparameter and filtered the test data for the adaptation. On the contrary, we used model trusting data, which is similar to the reliable samples but hypothetical. Considering that the reliable samples have low entropy which leads to near one-hot prediction, the observation from eata can be extended that it is crucial to adapt model with test data which gives near one-hot prediction (model trusting data). The primary goal of PIF is to prevent performance degradation on model-trusting data, rather than adapting the model with such data.
>
> Addressing your concern, it might seem puzzling why we introduced the hypothetical concept of 'model-trusting data' rather than utilizing the more concrete concept of 'reliable data.' This is because of the online-training nature of TTA. We cannot retrieve previous train/test data. Deciding whether to use the current data for adaptation is feasible in online training, but preventing performance degradation with any well-aligned data, without retrieval, is impossible. Therefore, we require a proxy to quantify the performance degradation. Fortunately, the parameters of the last layer (prototype) serve as a reliable approximation of the model-trusting data. Thus, we proposed a regularizer that quantifies the degradation of the prototype (a proxy for model-trusting data) through the influence function.
>
>
> **W2.** The experimental performance seems not prominent. And for different weighting strategies, only hard weight works fine. While the improvements are limited, the method introduces two extra hyperparameters, which raises concerns about whether using this PIF regularizer is a practical option.
>
> **A.** PIF consistently enhance the performance of all methods. Additionally, the weighting strategy employed in designing the PIF regularizer was a crucial design choice due to the absence of labels in the test data. Experimentally, aligning prototypes and gradients corresponding to hard pseudo-labels proved to yield the best performance compared to aligning all prototypes and gradients. However, the soft weighting strategy also demonstrated performance on par with or better than existing methods.
> PIF employs two hyperparameters, but it is not sensitive to their values. Particularly, in cases like CIFAR-10, good performance was observed without the use of decay hyperparameters. In scenarios such as ImageNet-C, even without employing decay techniques, applying PIF to TENT resulted in a 1.4% performance improvement, and applying it to PL resulted in a 2.7% performance improvement.
>
>
> **Q1.** Why didn't the authors compare SAR with LayerNorm and GroupNorm? As shown in SAR, SAR works much better using LN and GN. If PIF cannot be used in LN and GN and also cannot beat the results, can you convince me why I need PIF? Or can the author figure out how PIF can be used in LN and GN cases?
>
> **A.** SAR is a method designed for dynamic settings. Furthermore, the best performance was achieved when applying PIF to EATA.
>
>
> **Q2.** If I understand the paper right, PIF is basically aligning test samples with prototypes in gradient space. In such a case, one natural question is have you tried aligning them in embedding space? How does it perform? If I can align the first-order derivative, can I align the second-order? Will it provide a better performance?
>
> **A.** As you correctly mentioned, PIF aligns prototypes and test data in the gradient space. Since we use prototypes as weights in the last layer, the alignment of prototypes and test data in the embedding space is already handled by existing methods such as EM loss or PL loss. Aligning them in the Hessian space is not feasible in TTA settings due to computational efficiency constraints.

---

> > ### Comment · Reviewer_bVft · 2023-11-20
> > **Thank you for your response**
> >
> > Dear authors,
> >
> > Thank you for your detailed response. After I check your response and revised paper, however, I decide not to raise my score. The reasons are listed as follows.
> >
> > 1. In your response to W1, you mentioned preventing performance degradations. I acknowledge that it is an important aspect. However, in such case, the experiments are limited. More experiments on long test-time adaptation trajectories should be done to verify this claim. (Major)
> > 2. For Q1, the answer `SAR is a method designed for dynamic settings` is vague and not convincing.
> > 3. For Q2, I believe the answer is satisfactory. However, the experimental explanations fail to cover this important point.
> >
> > Overall, I think the overall method is clear, but the motivations and necessity of the methods and the corresponding experimental justifications are still limited.

---

### Official Review · Reviewer_nQF9 · 2023-11-01

**Soundness:** 3 good
**Presentation:** 3 good
**Contribution:** 3 good
**Rating:** 5
**Confidence:** 3

**Summary:**

This paper proposes utilizing the influence function for test-time adaptation. Since large time and computation complexity is required for influence function calculation, and since it is not adequate for using large amount of time and computation for test time adaptation regarding the problem's property, it proposes utilizing the weight of the model's last layer as prototype features.

**Strengths:**

- This paper suggests a scalable prototypical influence function which is adequate for test time adaptation.

**Weaknesses:**

- Since there is no label for test dataset, the test loss is often utiilized as entropy. Although I agree on that it is difficult to suggest new type of loss, I still think we cannot prove the learning direction is correct when trained with this type of loss. It could mean the influence calculated from the possibly wrong loss is also wrong.

**Questions:**

- I understood the weight vector of last layer may serve as a prototype of each class when there is no domain shift. However, if there is domain shift, can it also work as a prototype as the feature of source and features of target are distributed differently (more than Figure 1)? Or can we quantify the difference?
- Can authors explain the scheme of random projection (which means approximations of the hessian to the identity) in more details? I think this trick is quite important for your method to reduce the computation complexity, yet it is not explained enough in the paper.
- Why authors utilize L_PIF as regularizer? What happen if we do not use L_TTA? Why it would fail?

---

> ### Comment · Reviewer_nQF9 · 2023-11-20
>
> Reading the reviews from other reviewers' comments, I am convinced and lower my score to 5.

---

> ### Author Response · Authors · 2023-11-20
> **Response to Reviewer nQF9**
>
> We thank you for the insightful comments and suggestions. We have addressed each of your questions below
>
> **W1.** Since there is no label for test dataset, the test loss is often utiilized as entropy. Although I agree on that it is difficult to suggest new type of loss, I still think we cannot prove the learning direction is correct when trained with this type of loss. It could mean the influence calculated from the possibly wrong loss is also wrong.
>
> **A.** It is true that PIF depends on the form of the loss. I agree on the uncertainty of EM loss. We have derived PIF loss only for conventionally used EM and CE losses, but if there is a better loss, we can derive the corresponding PIF loss for it.
>
> **Q1.** I understood the weight vector of last layer may serve as a prototype of each class when there is no domain shift. However, if there is domain shift, can it also work as a prototype as the feature of source and features of target are distributed differently (more than Figure 1)? Or can we quantify the difference?
>
> **A.** We defined the prototype for model-trusting data. Model trusting data is defined as virtual data producing near one-hot predictions. To achieve near one-hot predictions, the features of the data should align well with the last layer weights. Therefore, we define the prototype of model-trusting data as the last layer weight. Thus, the definition of the prototype itself is not related to the shift between source and target distributions.
>
> **Q2.** Can authors explain the scheme of random projection (which means approximations of the hessian to the identity) in more details? I think this trick is quite important for your method to reduce the computation complexity, yet it is not explained enough in the paper.
>
> **A.** he Hessian inverse is a factor that determines how much weighting to give between two gradients. However, due to the nature of PIF, which requires calculating IF during training, it is infeasible to compute the exact Hessian inverse. Therefore, we used the random projection, an approximation technique used in conventional IF, to set the Hessian to identity and perform calculations.
>
> **Q3.** Why authors utilize L_PIF as regularizer? What happen if we do not use L_TTA? Why it would fail?
>
> **A.** The PIF derivation process itself regularizes to ensure positive impact on the model-trusting data when adapting test data with the original loss. Therefore, it is used as a regularizer. When using only PIF loss, it was effective, especially in small datasets like CIFAR-10-C, but had limitations for larger datasets like ImageNet-C.

---

### Official Review · Reviewer_hjny · 2023-11-01

**Soundness:** 2 fair
**Presentation:** 2 fair
**Contribution:** 3 good
**Rating:** 5
**Confidence:** 3

**Summary:**

The paper focuses on the test-time adaptation (TTA) task using the influence function. The proposed method aims to minimize the influence of test data on the loss of model-trusting data, which is approximated by class prototypes derived from the weights of the last layer in the classifier. Since the evaluation of influence functions is often computationally expensive and TTA is subject to resource constraints, two approximations are employed: parameter restriction and random projections. In the experiments, the proposed method consistently shows superior performance.

**Strengths:**

Introducing influence functions to TTA is reasonalbe, and the approach of combining influence function and TTA is not straightforward. Therefore, the proposed method could be a valuable contribution to the TTA community. The manuscript is well-written and easy to follow.

**Weaknesses:**

1. Additional support for influence function approximations

* The paper uses two approximation strategies for the influence function: parameter restriction and random projections, following previous work. Since this is the first application of the influence function in TTA, it is important to provide empirical evidence of the validity and impact of each approximation technique in TTA. A comparison with the exact influence function values, possibly by evaluating similarity or TTA performance, would be informative.

* The claim that normalization of gradients reduces error and improves stability needs stronger support. Direct comparisons of the influence function values would be more convincing.

2. Time complexity and memory requirement

* While approximations are applied to the influence function, it would be beneficial to discuss the time and memory requirements. A comparison of wall clock time and memory usage between the baseline and the proposed method could provide insight into the computational requirements.

3. Sensitivity of hyperparameters

* They only provide the ablation study on $\gamma$ for one setting of CIFAR-10-C. They mentioned that "sustained robust performance of the hyperparameter beyond a certain threshold". However, the optimal choice of $\gamma$ provided in Table 5 varies, and the performance gap is quite small for some settings. It would be better to provide the sensitivity experiments on the other settings and datasets.

* Also, they did not provide the sensitivity results for $\beta$. It would be great to provide them.

4. Presentation issues

* The full name of 'PIF' should be included in the abstract.
* It would be helpful to move the paragraph related to Figure 1 from the Introduction section to the Proposed Methods section. This experimental evidence of the prototypes would be better provided after a detailed explanation of the model confidence data, making it easier to understand.
* In Eq. (6), the right-hand side may need to be divided by $M$ because $\mathcal{L}(\mathcal{M})$ is the average loss on $\mathcal{M}$.
* Provide captions for Figure 3, and make sure that the basic experimental settings, such as the dataset, are described in the manuscript for clarity.

**Questions:**

Please answer the questions in the Weaknesses section.

---

> ### Author Response · Authors · 2023-11-20
> **Response to Reviewer hjny**
>
> We thank you for the insightful comments and suggestions. We have addressed each of your questions below.
>
> **W1.** Additional support for influence function approximations.
>
> **A.** The influence function was initially developed as a metric to understand black-box models, applicable after completing model training. For accurate influence function calculation, the inverse Hessian for the entire model parameter is required. However, as we measure the influence of test data on prototype loss at each iteration and raise its value, exact influence function computation is infeasible.
>  The inverse Hessian is typically used for weighting between two gradients. However, our use of random projection techniques results in the Hessian becoming an identity matrix. As a result, the scale has a more significant impact on the influence function value than the direction of the gradient. To address this, we normalize the two gradients when approximating the influence function. While the exact influence function values are not comparable due to the nature of PIF, comparing performance with and without normalization, as shown in Figure 3 of the paper, demonstrates the impact.
>
> **W2.** Time complexity and memory requirement
>
> **A.** Measuring time and memory usage for CIFAR-10 with Gaussian noise data, TENT required 12.50 seconds and 5250 MiB, while TENT+PIF required 20.30 seconds and 7660 MiB. Both time and memory increased, but the error rate decreased by 1.7%.
>
> **W3.** Sensitivity of hyperparameters
>
> **A.** For ImageNet-C dataset with TENT+PIF, fixing gamma at 150 and varying beta produced the following error rates:
> | beta       | 0    | 0.5  | 1    | 1.5  | 2    | 3    | 4    | 5    | 6    |
> |------------|------|------|------|------|------|------|------|------|------|
> | error rate | 64.8 | 58.4 | 55.9 | 55.3 | 55.3 | 55.6 | 55.9 | 56.2 | 56.3 |
>
> For ImageNet-C dataset with EATA+PIF, fixing beta at 2 and varying gamma produced the following error rates:
> | gamma      | 10    | 20    | 30    | 40    | 50    | 60    | 70    | 80    | 90    | 100   |
> |------------|-------|-------|-------|-------|-------|-------|-------|-------|-------|-------|
> | error_rate | 51.74 | 51.51 | 51.38 | 51.35 | 51.37 | 51.43 | 51.51 | 51.62 | 51.81 | 52.03 |
>
> **W4.** Presentation issues
>
> **A.** Currently, the revised portions are delineated in red in the manuscript

---

> > ### Comment · Reviewer_hjny · 2023-11-21
> >
> > Thank you for the response. Based on the authors' response and the reviews of other reviewers, I still have the following concerns.
> >
> > W1. Additional support for influence function approximations
> >
> > In the authors' response to another reviewer, the authors said that their objective is to maximize the influence function, so the exact calculation is not necessary. However, I think it is necessary to at least show that an approximation can maximize the exact value of the influence function. For example, by showing that the estimated value is the lower bound of the true value, by showing the gap between the two, or by experimentally showing the maximization of the true value.
> >
> > W2. Time complexity and memory requirement
> >
> > Thank you for providing the information. I think these results and the related discussion should be included in the revised version.
> > Also, it would be nice to see the better performance of the proposed model for the same limited total training time.
> >
> > W3. Sensitivity of hyperparameters
> >
> > As other reviewers have pointed out, I think it is a problem that the scale of the hyperparameters varies depending on the datasets and baseline models. I think the authors should provide guidelines on how to set the hyperparameters. Also, I would like to see a more detailed description of the hyperparameter selection process, such as which validation set the authors used for the greedy search.

---

### Meta-Review · Area_Chair_SNb8 · 2023-11-26

**Metareview:**

The paper studies the problem of test-time adaptation (TTA). The authors proposed to minimize the influence of test data on the loss of model-trusting data, which is approximated by class prototypes derived from the weights of the last layer in the classifier. Experiments show that the proposed method consistently achieves superior performance.

**Strengths**

- Introducing influence functions to TTA is reasonable and novel.

- The paper is well-written and easy to follow.

- The performance is good compared to existing methods.

**Weaknesses**

- Lack of additional support for influence function approximations.
- Time complexity and memory requirement.
- The proposed method is sensitive of hyperparameters.
- More experiments on long test-time adaptation trajectories should be done to verify this claim.
- Lack of explanations, including assumption behind weight prototypes and explanation of weighting strategy.

In sum, although the proposed method is novel in TTA and the paper is well-written, it still lacks of sufficient experiments and explanations to support the advantage of the proposed method. Also, the complexity and sensitivity of hyper-parameter limit the contributions of the proposed method. All the reviewers make a consistent decision that the current version of this paper does not meet the requirement of ICLR. The AC thus recommends reject.

**Justification For Why Not Higher Score:**

The paper should be largely improver in terms of more experiments and more clear explanations to support the advantage and effectiveness of the proposed method.

**Justification For Why Not Lower Score:**

N/A

---

### Decision · Program_Chairs · 2024-01-16

Reject